# Scalp Microbiome and Sebum Composition in Japanese Male Individuals with and without Androgenetic Alopecia

**DOI:** 10.3390/microorganisms9102132

**Published:** 2021-10-11

**Authors:** Kazuhiro Suzuki, Mizuna Inoue, Otomi Cho, Rumiko Mizutani, Yuri Shimizu, Tohru Nagahama, Takashi Sugita

**Affiliations:** 1Department of Microbiology, Meiji Pharmaceutical University, Kiyose 204-8588, Japan; kazuh-suzuki@taisho.co.jp (K.S.); chootomi@my-pharm.ac.jp (O.C.); 2Research & Development Headquarters Self-Medication, Taisho Pharmaceutical Co., Ltd., Saitama 331-9530, Japan; miz-inoue@taisho.co.jp (M.I.); r-mizutani@taisho.co.jp (R.M.); y-setoyama@taisho.co.jp (Y.S.); t-nagahama@taisho.co.jp (T.N.)

**Keywords:** androgenetic alopecia, scalp, microbiome, sebum composition, *Malassezia*

## Abstract

The skin microbiome and sebum may be associated with inflammation-related diseases of the scalp. To assess the pathogenesis and progression of androgenetic alopecia (AGA), we analyzed the composition of sebum and the bacterial and fungal microbiomes of the scalps of 118 Japanese male individuals with and without AGA, then discussed their roles in the pathogenesis of AGA. Sebum triglyceride and palmitic acid contents were higher in the AGA group than in the non-AGA group. *Malassezia restricta*, a lipophilic fungus that consumes palmitic acid, was abundant on the scalps of patients with AGA. *Cutibacterium*, *Corynebacterium*, and *Staphylococcus* were the most common genera in both groups, and patients with AGA exhibited scalp dysbiosis (increased abundance of *Cutibacterium* and decreased abundance of *Corynebacterium*). Our findings suggest that both sebum and the bacterial and fungal microbiomes of the scalp may be involved in the development of AGA.

## 1. Introduction

In androgenetic alopecia (AGA), the anagen becomes shorter and number of hair follicles in telogen increases during repeated hair cycles. Clinically, the hair on the frontal and parietal areas of the head becomes vellus hair, which is thinner and shorter, and finally there is no hair on the scalp [1,2,3]. In Japanese males, AGA becomes evident in the late 20s to 30s, then gradually progresses to completion after 40 years of age. The average incidence of AGA is approximately 30% in Japanese males of all ages [4]. Genetic predisposition and hormonal changes are involved in AGA pathogenesis [5]; for instance, a genome-wide association study identified 624 genomic sites related to AGA [6]. Studies on monozygotic and dizygotic twins have shown that genetic factors are strong [7,8], while studies on monozygotic twins have shown that internal and external factors affect alopecia [9], suggesting the involvement of factors other than heredity. Environmental factors affecting the scalp include sebum and microorganisms.

The sebum secreted by the scalp is a mixture of triglycerides (TGs), diglycerides (DGs), free fatty acids, squalenes, cholesterol esters, wax esters, and cholesterol [10]. Sebum secretion increases from puberty, peaks at 15 to 35 years of age, and declines continuously thereafter [11].

The human body is covered with various microorganisms, which constitute a microbial society termed a “microbiome.” *Cutibacterium acnes* and *Staphylococcus* species (bacteria) and *Malassezia* fungi predominate on the scalp [12,13]. Increased colonization by *Malassezia* and *Staphylococcus* may exacerbate dandruff production [12]. The involvement of the microbiota in scalp hypersensitivity and alopecia has been investigated. In a hypersensitive scalp, colonization by the lipophilic *C. acnes* increases with increasing sebum secretion, which disrupts the skin barrier [14]. *C. acnes*, *Staphylococcus*, and *Malassezia* have been implicated in AGA [15,16] and in dandruff production [17,18]. *Malassezia* and *C. acnes* lipases hydrolyze sebum into TGs, which are then hydrolyzed into glycerin and free fatty acids [19]. Moreover, *C. acnes* promotes lipogenesis in sebaceous glands [20].

Although sebum and the skin microbiome are presumed to be associated with inflammation of the scalp, no comprehensive analysis of the scalp microbiome or sebum has been performed in patients with AGA.

In this study, we analyzed the bacterial and fungal microbiomes and sebum composition of the scalps of Japanese male individuals with and without AGA, then discussed their roles in the pathogenesis of AGA.

## 2. Materials and Methods

### 2.1. Subject Recruitment

Scalp sebum and microbiome samples were collected from 55 Japanese males with AGA (16 in their 30s, 19 in their 40s, and 20 in their 50s), and 63 healthy individuals (without AGA [controls]; 12 in their 20s, 19 in their 30s, 17 in their 40s, and 15 in their 50s) at the Tokyo Center Clinic and AGA diagnosed (based on the Ogata scale) by expert physicians [4,21]. Patients with skin diseases other than AGA, patients who used drugs to encourage hair regrowth or to delay the progression of alopecia, and patients who applied drugs or quasi-drugs to the scalp (e.g., shampoos or hair-growth products containing antimicrobial agents) were excluded. All subjects washed their hair two nights prior to specimen collection; from that night until the day of collection, they were prohibited from washing or wetting their hair, rubbing their heads with sweat-wiping sheets or towels, using non-rinsing shampoos or hair dressings, causing extreme sweating (to the extent that it affected the head), and engaging in significant changes in daily life (e.g., commencement of strenuous exercise).

The protocols were reviewed and approved by the Institutional Review Board of the Japan Conference of Clinical Research (approval number: 229). Written informed consent was obtained from each subject.

### 2.2. Sample Collection

All heads were checked; the samples were collected principally from the parietal areas of both groups, especially from areas of hair loss in the AGA group. If sweating was observed at the time of specimen collection, the specimen was air-dried using a hair dryer (delivering cold air). Scalp microbiota and sebum were collected from different locations, but the two sites were chosen to be as close as possible while avoiding overlap.

To collect scalp microbiota, the hair was parted with a rubber-gloved hand to expose the scalp, and a swab (BD Falcon SWUBE Single Polyester Swab, Screw Cap, NJ, USA; dipped in a solution of 20 mM Tris [pH 8], 2 mM ethylenediaminetetraacetic acid, and 1.2% [*v*/*v*] Triton X-100) was used to vigorously rub the skin in the parietal region of the head for 1 min, moving 15 times in both the x- and y-directions over an area of 3 cm^2^. DNA was extracted as described previously [22], then precipitated with ethanol using Ethachinmate (Nippon Gene, Toyama, Japan) as a precipitation activator, in accordance with the manufacturer’s instructions. Unused swabs and reagents served as negative controls.

To collect sebum, the hair was parted with a spatula to expose the scalp. Degreased cotton had previously been washed with hexane for 16 h in a Soxhlet extractor and dried at 60 °C for 3 h. A piece of this degreased cotton was clipped with tweezers and pressed lightly against the scalp, then moved gently back and forth, and left and right, over an area of approximately 5 cm × 2 cm (to collect sebum). This process was performed approximately five times. All samples were frozen until use.

### 2.3. Analysis of the Skin Microbiome

The V1–V3 regions of bacterial 16S rRNA genes and the D1–D2 regions of fungal 28S rRNA genes were amplified by PCR using primers 27F (5′-AGAGTTTGATCCTGGCTCAG-3′) and 534R (5′-ATTACCGCGGCTGCTGG-3′) for bacteria and NL1 (5′-GCATATCAATAAGCGGAGGAAAAG-3′) and NL4 (5′-GGTCCGTGTTTCAAGACGG-3′) for fungi [23]. Pooled amplicons were prepared for sequencing using a Nextera XT DNA Library Preparation Kit (Illumina, San Diego, CA, USA) and sequenced on a MiSeq platform using MiSeq version 3 Reagent Kits (Illumina), in accordance with the manufacturer’s instructions. Microbiome analysis was performed with the aid of the QIIME 2 pipeline (https://qiime2.org, accessed on 4 June 2021); additional analyses and visualizations were conducted using R, version 4.1.0 (https://www.R-project.org, accessed on 4 June 2021). The sequence data were demultiplexed and quality-filtered using the q2-demux plugin, then denoised with DADA2 [24], to identify all observed amplicon sequence variants [25]. These variants were aligned by employing mafft [26]; fasttree was used to construct phylogenies [27]. Alpha-diversity metrics, beta diversity metrics, and principal coordinates analysis were constructed using q2-diversity data. Taxonomy was assigned using Silva [28] version 138 (99% of the full-length sequences of OTUs) for bacterial 16S sequences and RDP (https://rdp.cme.msu.edu, accessed on 4 June 2021) for fungal 28S sequences. Because fungal taxonomy is under revision, the latest fungal taxa were downloaded from MycoBank (https://www.mycobank.org, accessed on 4 June 2021).

The *Malassezia* colonization level was quantified using a real-time PCR assay employing a TaqMan probe, in accordance with the method established by Sugita et al. [29]. Amplification and detection were performed using an ABI PRISM 7500 platform (Applied Biosystems, Foster City, CA, USA).

### 2.4. Analysis of Scalp Sebum

A 6-mL mixture of chloroform and methanol (2:1 *v*/*v*) was placed in a test tube containing degreased cotton covered with sebum; the tube was filled with nitrogen gas and shaken for 30 min. The resulting solution was stored in an airtight container in an ultra-low temperature freezer (Nihon Freezer Co., Ltd., Tokyo, Japan) (−100 to −60 °C) prior to the measurement of free fatty acids, DGs, TGs, squalenes, free cholesterol, cholesterol esters, and waxes.

#### 2.4.1. Free Fatty Acid Analysis

The Supelco 37 component FAME Mix (Sigma-Aldrich Corp., St. Louis, MO, USA) served as the standard during free fatty acid analysis by means of gas chromatography-mass spectrometry (GCMS). We used a gas chromatograph (GC-2010Plus), a mass spectrometer (GCMS-QP2010Ultra), a GCMS auto-sampler (AOC-20s), a GCMS auto-injector (AOC-20i), and GCMSsolution software (all from Shimadzu Corp., Kyoto, Japan). The extract solution was fractionated in terms of free fatty acids using solid phase cartridges. After methyl esterification, each fatty acid was analyzed (via GCMS) as its fatty acid methyl ester. The mass range was mass-to-charge ratio (*m/z*) 33–450. Saturated and branched-chain fatty acids, monounsaturated fatty acids, and polyunsaturated fatty acids were measured using mass spectrum base peaks (*m*/*z* peaks) of 74, 55 and 81, respectively, in the mass chromatogram.

#### 2.4.2. Glyceride Analysis

Monopalmitin (Combi-Blocks, Inc., San Diego, CA, USA), dipalmitin (Accustandard, Inc., New Haven, CT, USA), and tripalmitin (Fujifilm Wako Pure Chemical Corp., Osaka, Japan) served as standards for glyceride analysis. The gas chromatograph, GC autosampler (AOC-20s), GC autoinjector (AOC-20i), and LabSolutions software from Shimadzu Corp. were used for measurements. The glyceride concentrations in sample solutions were calculated based on the peak areas of 50 μg/mL of a mixed standard solution of palmitins. Because the peaks of diglycerides overlapped with the peaks of wax esters, the diglyceride concentrations were determined by subtracting the concentrations of wax esters from the concentrations of diglycerides.

#### 2.4.3. Free Cholesterol and Squalene Analysis

Cholesterol (Fujifilm Wako Pure Chemical Corp., Osaka, Japan), squalene (Tokyo Chemical Industry Co., Ltd., Tokyo, Japan), cholesterol myristate (Tokyo Chemical Industry Co.), cholesterol palmitate (Tokyo Chemical Industry Co.), and cholesterol stearate (Sigma-Aldrich Corp.) served as standards during free cholesterol and squalene analysis. High-performance liquid chromatography (HPLC; 30A (Nexera X2) series), tandem mass spectrometry (MS/MS; LCMS-8060), and Labsolutions software version 5.86 from Shimadzu Corp. were used for measurements. A calibration curve was prepared from a linear regression equation (using the least-squares method) of the peak area ratios of the calibration curve samples. The sample values were calculated by fitting the peak area ratios of the sample solutions and recovered samples to the calibration curve. If lipids other than cholesterol myristate, cholesterol palmitate, and cholesterol stearate were detected, the measured values were calculated using the calibration curves of those standards, which exhibited adjacent retention times.

#### 2.4.4. Wax Ester Analysis

Dodecyl stearate (Tokyo Chemical Industry Co., Ltd., Tokyo, Japan) served as the standard during analysis of wax esters. An HPLC platform (30A (Nexera X2) series), an MS/MS unit (LCMS-8060 series), and Labsolutions software version 5.86 from Shimadzu Corp. were used for measurements. A calibration curve was prepared from a linear regression equation (using the least-squares method) of the peak area ratios of the calibration curve samples. The sample values were calculated by fitting the peak area ratios of the sample solutions and recovered samples to the calibration curve. If wax esters other than dodecyl stearate were detected, the measured values were calculated using the calibration curve of dodecyl stearate.

### 2.5. Statistics

We used Student’s *t*-test to compare means; we also calculated Pearson product-moment correlation coefficients.

## 3. Results

### 3.1. Skin Microbiome Analysis

The bacterial and fungal microbiomes of the scalps of 55 Japanese males with AGA and 63 Japanese males without AGA were analyzed. *Cutibacterium* predominated in the bacterial microbiome at all ages and scalp sites in the AGA and non-AGA groups, followed by *Corynebacterium*, *Staphylococcus*. These three taxa accounted for >90% of all bacteria in both groups. *Cutibacterium* (51.2% ± 31.2% (AGA group) vs. 41.5% ± 27.0% (non-AGA group), *p* > 0.05) and *Staphylococcus* (18.7% ± 23.6% (AGA group) vs. 12.0% ± 14.1% (non-AGA group), *p* > 0.05) were more abundant whereas *Corynebacterium* (24.0% ± 28.3% (AGA group) vs. 37.6% ± 30.4% (non-AGA group), *p* < 0.05), which was less abundant in the AGA group than in the non-AGA group (Figure 1, Appendix A). The Shannon diversity index indicated no bacterial variation between the two groups (Figure 2).

In terms of the fungal microbiome, taxa with > 5% relative abundance are shown in Figure 1. *Malassezia restricta* predominated at all ages in both AGA and non-AGA groups (64.7% ± 37.5% (AGA group) vs. 44.6% ± 39.0% (non-AGA group), *p* < 0.05), followed by *Aureobasidium* [6.6% ± 20.0% (AGA group) vs. 8.4% ± 22.7% (non-AGA group), *p* > 0.05] and *Rhodotorula mucilaginosa* (9.6% ± 20.3% (AGA group) vs. 7.5% ± 17.3% (non-AGA group), *p* > 0.05) (Figure 1). Irrespective of age, *M. restricta* was more abundant in the AGA group than in the non-AGA group. The Shannon diversity index indicated no fungal variation between the two groups (Figure 2). qPCR revealed that the level of *Malassezia* colonization was greater in the AGA group than in the non-AGA group at any age, and *Malassezia* was significantly more predominant in subjects in their 50s than in subjects in their 30s or 40s in the AGA group (Figure 3, *p* < 0.05). The ratio of the number of *M. restricta* and *M. globosa* reads was calculated using the number of reads for all *Malassezia* species as 100%. For the genus *Malassezia*, the ratio of *M. restricta* to the overall *Malassezia* level increased with age, whereas the ratio of *M. globosa* decreased with age (Figure 4).

### 3.2. Sebum Composition

Sebum consists of TGs, DGs, free fatty acids, squalenes, cholesterol esters, free cholesterol, and wax ester (Table 1). TGs were most abundant, accounting for ~50% of the total lipid content, followed by free fatty acids, DGs, and squalenes. The TG ratio was significantly higher in the AGA group than in the non-AGA group (*p* < 0.05), whereas the free fatty acid, squalene, and free cholesterol ratios were significantly lower in the AGA group than in the non-AGA group (*p* < 0.05). The TG and free fatty acid ratios in subjects in their 30s differed between the two groups (Appendix A).

Because free fatty acids are produced by hydrolysis of TGs and DGs, we expected to find a correlation between the ratios of various free fatty acids and TGs. The correlation coefficients for these ratios were −0.96 in the AGA group and −0.95 in the non-AGA group, confirming that the sebum composition had been determined correctly; the percentages of sebum relative to the sum of free fatty acids, TGs, and DGs were 82.3% ± 3.0% in the AGA group and 80.6% ± 3.4% in the non-AGA group.

### 3.3. Free Fatty Acids

The results of measurements above the lower limit of quantification (3.0 μg/mL) of 47 free fatty acids in scalp sebum were analyzed (Table 2). Our use of GC-MS precluded identification of the position of the unsaturated bond or the branching position of the carbon chain in fatty acids marked with (*). In descending order, we detected C16:0 (palmitic acid), C16:1_1 (estimated to be sapienic acid), C14:0 (myristic acid), C18:1 (structure undetermined, but estimated to be oleic acid), and C15:0 (pentadecanoic acid). These five free fatty acids accounted for >70% of the total content. Only the C16:0 (palmitic acid) level was significantly higher in the AGA group (*p* < 0.05).

### 3.4. Relationship between the Scalp Microbiome and Sebum Composition

The relationship between the relative abundance of scalp microbiota (*Cutibacterium*, *Corynebacterium*, *Staphylococcus*, *Bradyrhizobium, M. restricta, M. globosa, Aureobasidium*, and *Rhodotorula mucilaginosa*) and sebum composition was analyzed. In the AGA group, the relative abundance of *Cutibacterium* was positively correlated with the ratio of TGs (correlation coefficient, *r* = 0.41) and negatively correlated with the ratio of free fatty acids (*r* = −0.47). By contrast, the relative abundance of *Corynebacterium* was negatively correlated with the ratio of TGs (*r* = −0.60) and positively correlated with the ratio of free fatty acids (*r* = 0.65) (Figure 5).

Because free fatty acids are produced by the hydrolysis of TGs and DGs, an increase in the ratio of TGs to DGs is associated with a decrease in the free fatty acid ratio. The abundance of *Cutibacterium* exhibited a positive correlation with the ratio of TGs to DGs and a negative correlation with the free fatty acid ratio. In the AGA group, a decrease in the abundance of *Corynebacterium* was associated with an increase in the abundance of *Cutibacterium* and was negatively correlated with the ratio of TGs to DGs.

## 4. Discussion

To determine the effects of the microbiome and sebum composition on the development of AGA, we compared the scalp microbiome and sebum composition between patients with AGA and healthy individuals. *Cutibacterium*, *Staphylococcus*, and *Corynebacterium* were the most common bacteria, while *Malassezia* was the most common fungus, on healthy scalps [12,13]. The scalp microbiome was similar between patients with dandruff and healthy subjects, but with different relative abundances of microorganisms. The abundance of *C. acnes* (compared to the abundance of *S. epidermidis*) increased on the scalps of patients with acne vulgaris [30] or dandruff [17], compared to healthy individuals. Similarly, *C. acnes* was more abundant than *Corynebacterium* on the skin of patients with psoriasis than on the skin of healthy individuals [31]. Therefore, a balance between skin microbes maintains skin homeostasis and an imbalance causes inflammation. *Cutibacterium* is the most important microbe for skin homeostasis and contributes to the biophysiological functions of the skin via lipid modulation, niche competition, and oxidative stress mitigation [32]. We observed increased abundances of *Cutibacterium* and *Staphylococcus* and a decreased abundance of *Corynebacterium* in the AGA group, compared to the non-AGA group (Figure 1, Appendix A).

Pinto et al. [33] found an increased abundance of *Cutibacterium* and a decreased abundance of *Staphylococcus* in Italian patients with alopecia areata (AA), compared to healthy controls. Their results and ours differ in terms of the abundances of *Staphylococcus* and *Corynebacterium*. Although this may be attributable to differences in ethnicity (Italians in their study and Japanese in our study) and sexes (both sexes in their study but men only in our study), both works suggest that an imbalance between the two species is involved in the development of AA and AGA.

*Cutibacterium* produces an antimicrobial thiopeptide, cutimycin, which modifies the microbiome composition of microenvironments, such as hair follicles [34]. The expression of cutimycin genes is increased if *C. acnes* is co-cultured with *Staphylococcus* and decreased if it is co-cultured with *Corynebacterium.* Triacylglycerol lipase produced by *C. acnes* hydrolyzes sebum into short-chain fatty acids (SCFAs), including acetate (C2), propionate (C3), and butyrate (C4) [32]. SCFAs are important for skin homeostasis and maintain a weakly acidic microenvironment. Conversely, porphyrins produced by *C. acnes* increase the production of inflammatory cytokines in keratinocytes, leading to skin inflammation [35]. *Cutibacterium* and photoactivated porphyrins in the pilosebaceous ducts increase oxidative tissue injury and cause follicular micro-inflammation [36]. The ability to produce porphyrins differs among *C. acnes* phylotypes [37]. It is, therefore, of interest to determine whether a particular phylotype is present on the scalps of AGA patients.

The bacterial microbiome composition of healthy skin differs depending on the body part, but the fungal microbiome is similar (*Malassezia* predominates) regardless of the body site, except for the sole of the foot [38]. *M. restricta* is the most abundant *Malassezia* species, followed by *M. globosa*. *M. restricta* and *M. globosa* are present at a 3:1 ratio on the face of healthy subjects [39]. By contrast, *M. restricta* predominates over *M. globosa* (9:1) on the skin of patients with dandruff and seborrheic dermatitis [40]. In addition, the colonization of *Malassezia* is sixfold higher at lesional sites than at non-lesional sites, and microbial diversity decreases at lesional sites. This is because lipophilic *Malassezia* become increasingly predominant as the amount of sebum increases. In patients with AGA, *Malassezia* colonization was approximately twofold higher than in subjects without AGA. We calculated the ratio of abundance of each fungus to the total number of *Malassezia* species identified. In the AGA group, the abundance of *M. restricta* increased with age (85.7% in 30s, 93.0 in 40s, and 95.6% in 50s), whereas the abundance of *M. globosa* decreased with age (14.0% in 30s, 4.2% in 40s, and 4.2% in 50s) (Figure 4). A higher number of *Malassezia* cells was detected in hair roots from the AGA group than from the non-AGA group. The microbes were embedded in the superficial hair root tissue of patients with AGA [16], suggesting that *Malassezia* cells may have a direct effect on hair roots.

TGs were the most abundant lipids on the scalps of Japanese males, accounting for approximately 50% of all lipids, followed by free fatty acids, DGs, and squalenes. A significantly higher TG ratio and lower free fatty acid, squalene, and free cholesterol ratios were observed in the AGA group, compared with the non-AGA group. Moreover, in the non-AGA group, free fatty acids tended to decrease, and TGs tended to increase in individuals in their 40s and 50s, compared to individuals in their 20s. In contrast, in the AGA group, the free fatty acid ratio was low, and the TG ratio was high in individuals in their 30s, similar to the trend observed in older non-AGA subjects. Therefore, the scalps of individuals with AGA show features of aging from a young age. In addition, sebum composition changed with advancing age before AGA onset in the non-AGA group and exhibited a pattern similar to that of the AGA group, suggesting that sebum composition changes may be one of the causes of AGA. The contents of the free fatty acids, C16:0 (palmitic acid), C16:1_1 (estimated to be sapienic acid), C14:0 (myristic acid), C18:1 (structure undetermined, but estimated to be oleic acid), and C15:0 (pentadecanoic acid) were high, accounting for about 70% of the total free fatty acids. Palmitic acid was significantly more abundant on the scalps of individuals with AGA than on the scalps of individuals without AGA. Fatty acids, such as palmitic acid and oleic acid, are generated by lipases produced by *Malassezia* and *Cutibacterium acnes*; these compounds lead to inflammation [41,42]. In addition, inflammation reduces the efficacy of hair loss treatment and chronic microinflammation of the scalp leads to hair loss [43], suggesting that palmitic acid induces hair loss by triggering inflammation [41]. In the present study, sebum and microbiome changes were observed on the scalp in the AGA group. These changes may increase inflammation, leading to AGA progression. There was no difference in the amount of scalp sebum produced by individuals in the two groups [44]. However, the hair volume was lower in patients with AGA; thus, sebum does not adhere to the hair and remains on the scalps of AGA patients. Therefore, AGA patients have more sebum on their scalp, higher abundances of lipophilic *Malassezia* and *Cutibacterium,* and increased TG content, compared to non-AGA individuals; the ratio of TGs increases accordingly. The palmitic acid content was significantly higher in the AGA group. This may be linked to metabolism of free fatty acids by lipophilic microbes to create a more hospitable environment. Furthermore, it is possible that changes in sebum quantity and composition may affect the hydration level in the stratum corneum, which in turn affects the scalp environment.

The proportion of palmitic acid, a saturated fatty acid, in the sebum free fatty acids was increased in the AGA group. Because *Malassezia* assimilates and consumes palmitic acid, a palmitic acid-rich environment is suitable for the growth of *Malassezia*. *Malassezia* also has satisfactory growth in palmitic acid-supplemented medium in vitro [45].

In conclusion, sebum production and scalp environment are affected by age, hormones [46], diet, lifestyle [47,48], stress [49], and ultraviolet radiation [50]. In addition, changes in sebum composition and/or the scalp microbiome may be involved in AGA development and progression. Further research is needed to understand whether modifications of the aforementioned factors may prevent or improve AGA.

## Figures and Tables

**Figure 1 microorganisms-09-02132-f001:**
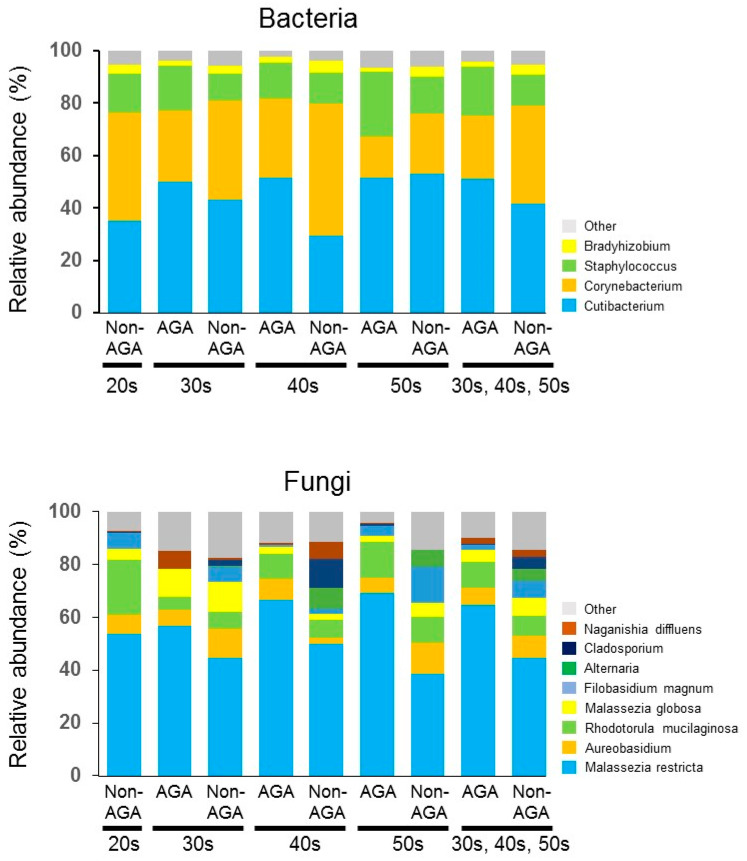
Mean relative abundances of the top four most abundant bacterial and fungal taxa (>5% abundance) in scalp samples from AGA and non-AGA groups by age group (20s, 30s, 40s, and 50s). There was a higher abundance of *Cutibacterium* and *Staphylococcus* and lower abundance of *Corynebacterium* in the AGA group than in the non-AGA group. In the fungal microbiome, *Malassezia restricta* predominated at all ages in the AGA and non-AGA groups.

**Figure 2 microorganisms-09-02132-f002:**
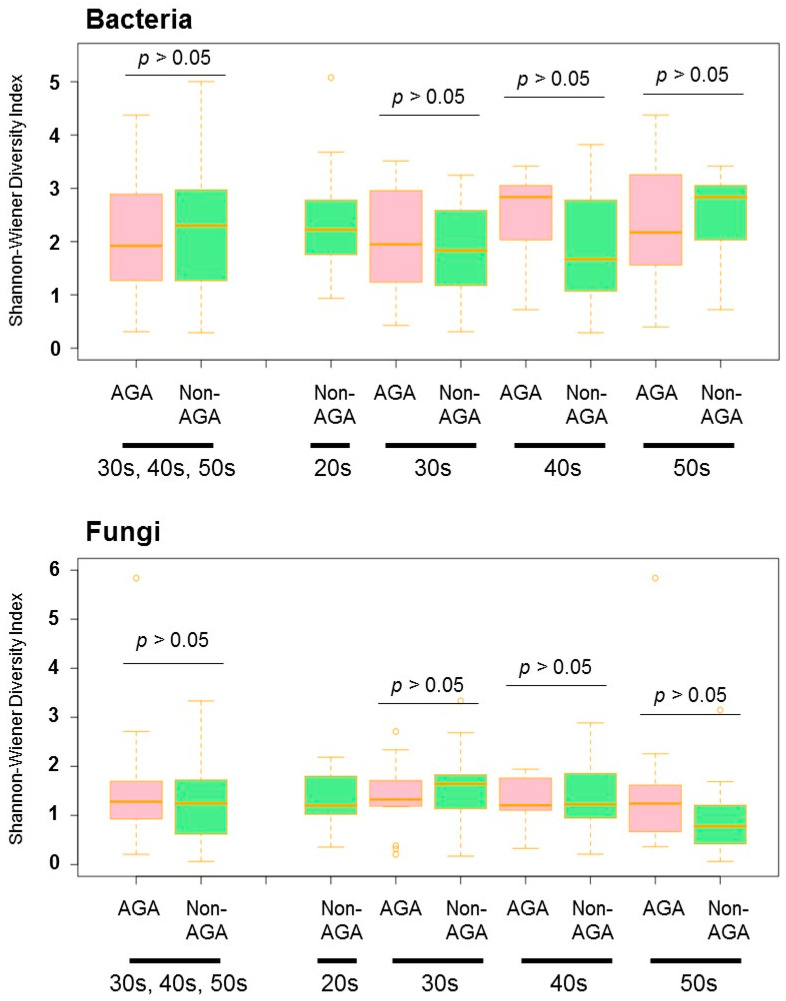
Shannon diversities of microbial communities among AGA and non-AGA subjects by age group (20s, 30s, 40s, and 50s).

**Figure 3 microorganisms-09-02132-f003:**
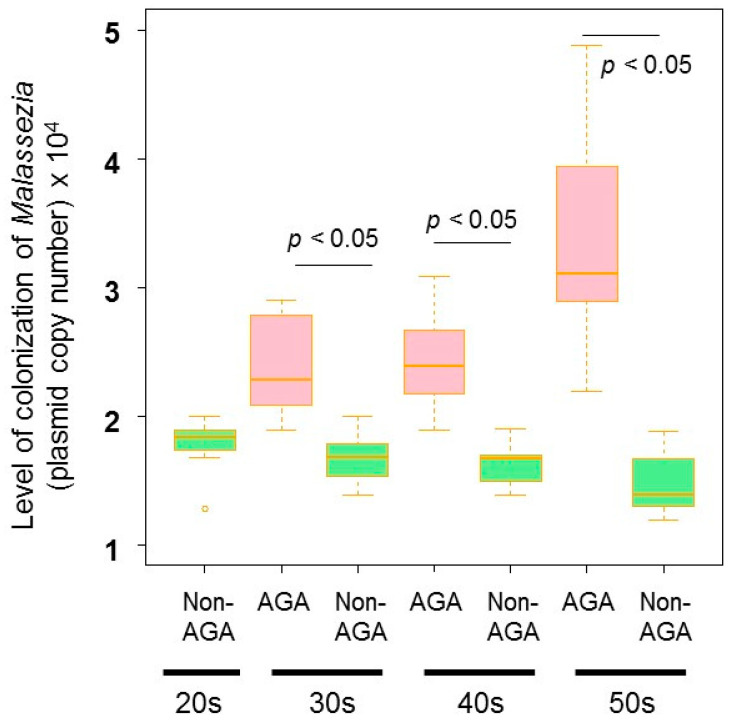
qPCR analysis of *Malassezia* colonization. There was a greater degree of *Malassezia* colonization in the AGA group than in the non-AGA group at all ages. The total *Malassezia* species colonization levels were determined via real-time PCR using a TaqMan probe.

**Figure 4 microorganisms-09-02132-f004:**
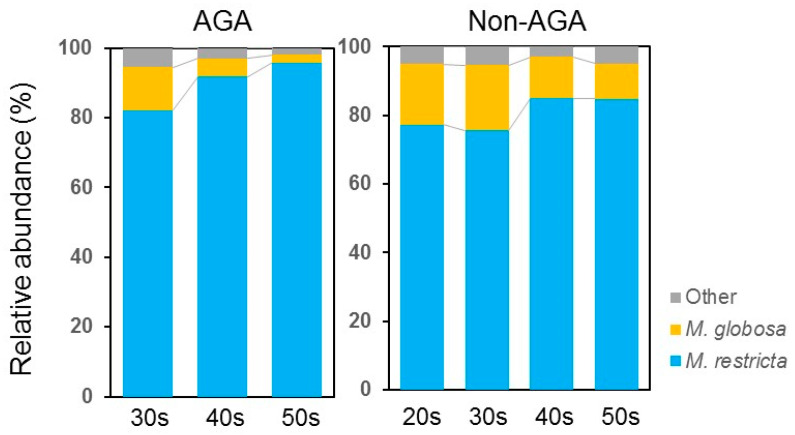
Ratio of *Malassezia restricta* to *M. globosa* by age. The ratio of the number of *M. restricta* and *M. globosa* reads was calculated using the number of reads for all *Malassezia* species as 100%.

**Figure 5 microorganisms-09-02132-f005:**
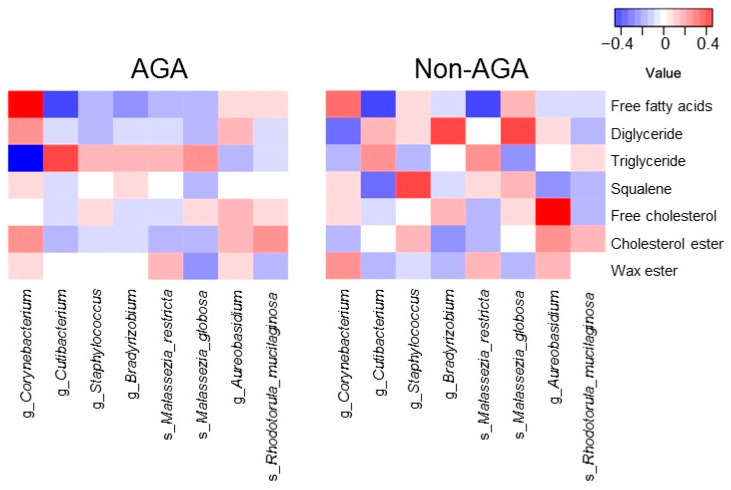
Heatmap of the predominant scalp microbiota and sebum composition.

**Table 1 microorganisms-09-02132-t001:** Composition (%) of sebum from the scalps of the androgenetic alopecia (AGA) and non-AGA groups.

Component	AGA Group	Non-AGA Group	*p*-Value
Triglyceride	54.4 ± 10.5	46.6 ± 10.7	0.0001
Diglyceride	14.3 ± 2.3	15.2 ± 2.7	0.0614
Free fatty acid	13.6 ± 7.5	18.7 ± 7.2	0.0002
Squalene	10.7 ± 2.0	12.1 ± 2.2	0.0004
Cholesterol ester	5.6 ± 2.2	5.8 ± 1.9	0.6741
Free cholesterol	0.8 ± 0.3	1.0 ± 0.5	0.0135
Wax ester	0.6 ± 0.5	0.5 ± 0.5	0.6810

Compositions are represented as means ± standard deviations. *p*-values were calculated using Student’s *t*-test.

**Table 2 microorganisms-09-02132-t002:** Composition (%) of free fatty acids in sebum of the AGA and non-AGA groups.

Type of Free Fatty Acids	AGA Group	Non-AGA Group	*p*-Value
C16:0	33.10 ± 5.15	31.35 ± 3.29	0.0276
(C16:1_1)	12.53 ± 3.08	12.29 ± 2.43	0.6380
C14:0	11.55 ± 1.89	11.11 ± 1.24	0.1381
(C18:1)	11.22 ± 2.40	11.13 ± 2.13	0.8455
C15:0	8.16 ± 1.72	8.36 ± 1.40	0.4889
C18:0	3.70 ± 1.39	3.48 ± 0.59	0.2562
C17:0	2.84 ± 0.67	2.93 ± 0.58	0.4427
(Branched C16:0_3)	1.79 ± 0.91	2.23 ± 0.93	0.0108
(Branched C14:0)	1.47 ± 0.85	1.87 ± 0.82	0.0113
(Branched C15:0_3)	1.31 ± 0.60	1.26 ± 0.38	0.5798
(Branched C16:1)	1.14 ± 0.74	1.40 ± 0.61	0.0414
(C17:1_2)	1.11 ± 0.44	1.19 ± 0.32	0.2561
(C16:1_2)	1.06 ± 0.46	1.11 ± 0.35	0.5138
(C14:1)	0.96 ± 0.48	0.99 ± 0.38	0.7238
C24:0	0.82 ± 0.45	0.80 ± 0.31	0.7859
C12:0	0.79 ± 0.47	0.78 ± 0.30	0.8865
(C17:1_3)	0.79 ± 0.40	0.84 ± 0.28	0.4200
(C15:1)	0.68 ± 0.34	0.70 ± 0.30	0.8021
C18:2n6t	0.61 ± 0.35	0.69 ± 0.29	0.1469
(Branched C18:1)	0.56 ± 0.52	0.75 ± 0.49	0.0425
(Branched C17:0_1)	0.52 ± 0.36	0.63 ± 0.27	0.0542
(Branched C17:0_2)	0.39 ± 0.30	0.43 ± 0.24	0.4769
C13:0	0.39 ± 0.32	0.50 ± 0.25	0.0333
(C17:1_1)	0.29 ± 0.28	0.32 ± 0.24	0.5850
(C18:2)	0.28 ± 0.26	0.33 ± 0.24	0.2331
C20:0	0.26 ± 0.27	0.35 ± 0.23	0.0674
C22:0	0.23 ± 0.26	0.31 ± 0.19	0.0483
(C20:2)	0.20 ± 0.24	0.23 ± 0.22	0.4610
(Branched C15:0_1)	0.19 ± 0.23	0.27 ± 0.22	0.0421
C18:2n6c	0.18 ± 0.23	0.31 ± 0.29	0.0097
(Branched C15:0_2)	0.16 ± 0.30	0.16 ± 0.21	0.9575
C18:1n9c	0.13 ± 0.25	0.07 ± 0.17	0.1856
Others_3	0.12 ± 0.19	0.18 ± 0.19	0.1365
Others_4	0.11 ± 0.20	0.18 ± 0.23	0.0964
(Branched C16:0_2)	0.09 ± 0.16	0.12 ± 0.17	0.3721
(C26:0)	0.07 ± 0.15	0.09 ± 0.14	0.5248
(Branched C25:0)	0.04 ± 0.13	0.04 ± 0.10	0.9482
(C19:0)	0.03 ± 0.08	0.07 ± 0.12	0.0629
(Branched C13:0_2)	0.03 ± 0.08	0.03 ± 0.07	0.9727
Others_1	0.03 ± 0.07	0.04 ± 0.09	0.2804
(Branched C16:0_1)	0.02 ± 0.05	0.02 ± 0.06	0.6507
(C25:0)	0.01 ± 0.05	0.03 ± 0.07	0.1634
(Branched C13:0_1)	0.01 ± 0.05	0.02 ± 0.07	0.4160
C23:0	0.01 ± 0.03	0.01 ± 0.03	0.7156
(Branched C15:1)	0.01 ± 0.03	0.00 ± 0.00	0.0624
Others_2	0.01 ± 0.03	0.00 ± 0.00	0.1292

The proportion of each free fatty acid was calculated by regarding the total free fatty acid level as 100%. The data are means ± standard deviations. *p*-values were calculated using Student’s *t*-test.

## Data Availability

Detailed data are provided in the Appendix A.

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
