# Peer review of "Scalp Microbiome and Sebum Composition in Japanese Male Individuals with and without Androgenetic Alopecia"

_microorganisms, 2021, doi:10.3390/microorganisms9102132_

Round 1

Reviewer 1 Report

The idea that follicular inflammation plays a role in AGA has been around for over 30 years (Kligman 1988) and it was noticed some years before that. Whether inflammation plays a causal role in the hair loss remains uncertain but understanding why it occurs could be important. It is pleasing therefore to see a study that has pursued this aspect of AGA through analysis of the microbiome and sebum composition. The results are novel and interesting but I think a rather more considered conclusion is needed. At issue is whether the differences observed between AGA and non-AGA scalp are cause or effect (or a bit of both).

  1. The microenvironments on non-bald and bald scalp are different – one has hair, the other does not. Could this influence the composition of the microbiome? The distribution of sebum, for example, is different. Could scalp surface humidity/hydration differ? Perhaps consider a different control group e.g. non-bald men who shave their scalps on a long-term basis.  
  2. If the change in sebum composition has a determining role in AGA why do hair follicles transplanted from the occipital scalp survive? – one would not expect the sebum composition to change as the original sebaceous glands are still in situ. I could understand how the mechanism proposed by the authors may have a modulating effect at a relatively late stage in the pathogenesis of AGA but am less convinced it is a key driver of events.

The discussion of the genetics of male AGA could be improved. The Gatherwright twin study (ref 7) only assessed identical twins, although it does not state how monozygosity was determined, and does not provide a heritability estimate. The twin studies by Nyholt et al (2003)and Rexbye et al (2005) both of which included dizygotic as well as monozygotic twins, estimated the heritability of male AGA at between about 80-95%. There are also several more recent studies of AGA genomics than the reference quoted (ref 6), e.g. the most recently published GWAS identified 624 genomic sites associated with AGA (Yap et al Nat Commun. 2018;9:5407).

Author Response

Reviewer 1

Thank you for your helpful comments and suggestions. We have modified the text accordingly.

Q1. The idea that follicular inflammation plays a role in AGA has been around for over 30 years (Kligman 1988) and it was noticed some years before that. Whether inflammation plays a causal role in the hair loss remains uncertain but understanding why it occurs could be important. It is pleasing therefore to see a study that has pursued this aspect of AGA through analysis of the microbiome and sebum composition. The results are novel and interesting but I think a rather more considered conclusion is needed. At issue is whether the differences observed between AGA and non-AGA scalp are cause or effect (or a bit of both).

Response

We agree that it is crucial to determine whether differences between AGA and non-AGA scalps are causes, consequences, or both of AGA. We believe that the differences between AGA and non-AGA scalps may reflect the development and progression of AGA. Although it is still unclear whether inflammation is the cause of hair loss, phenomenologically, it may be one of the causes. The incidence of AGA increases with age. Additionally, with increasing age, the sebum composition in the non-AGA group was similar to that in the AGA group. Therefore, sebum composition changes appeared before AGA onset, suggesting that sebum changes may be one of the causes of AGA. Triglycerides are hydrolyzed into free fatty acid by lipase produced by lipophilic microorganisms such as Malassezia spp. and Cutibacterium acnes. Some free fatty acids can induce inflammation. In addition, because free fatty acids are used as an energy source by lipophilic microorganisms, an increase in triglycerides may provide a favorable environment for these microorganisms. Because lipophilic microorganisms induce inflammation and chronic scalp inflammation causes alopecia, changes in the microbiome may be a cause of AGA. In this study, sebum and microbiome changes were observed on AGA scalps, with these changes potentially causing inflammation and worsening AGA. Therefore, sebum and microbiome changes may lead to AGA development and progression.

Based on your suggestion, we have added the following sentences to the Discussion:

In addition, sebum composition changed with advancing age before AGA onset in the non-AGA group and exhibited a pattern similar to that of the AGA group, suggesting that sebum composition changes may be one of the causes of AGA (Lines 326–329).

In the present study, sebum and microbiome changes were observed on the scalp in the AGA group. These changes may increase inflammation, leading to AGA progression (Lines 338–340).

Additionally, we have modified the following sentences in the Conclusion:

In conclusion, sebum production and the scalp environment are affected by age, hormones [46], diet, lifestyle [47, 48], stress [49], and ultraviolet radiation [50]. In addition, changes in sebum composition and/or the scalp microbiome may be involved in AGA development and progression. Further research is needed to understand whether modifications of the aforementioned factors may prevent or improve AGA (the word “progression” has been added to Lines 357).

Q2. The microenvironments on non-bald and bald scalp are different – one has hair, the other does not. Could this influence the composition of the microbiome? The distribution of sebum, for example, is different. Could scalp surface humidity/hydration differ? Perhaps consider a different control group e.g. non-bald men who shave their scalps on a long-term basis.

Response

The presence or absence of hair changes the microbiome and sebum composition (please see the response to Q1). These changes may affect AGA development and progression. As you commented, humidity and moisture may also affect the scalp microenvironment.

However, Chanprapaph et al. (Clin Interv Aging 2021 May 10;16:781–87. doi: 10.2147/CIA.S310178) included an Asian population and reported a lower hydration level for the scalp stratum corneum in the parietal compared to occipital region in the AGA group. However, this difference was not observed in the healthy group, suggesting that the scalp stratum corneum hydration level is affected by the occlusion of the scalp surface by sebum and the thickness of the stratum corneum. We believe that the difference in moisture level is secondary to differences in the quantity and composition of sebum and the thickness of the stratum corneum.

Based on your suggestion, we have added the following sentences to the Discussion:

Furthermore, it is possible that changes in sebum quantity and composition may affect the hydration level in the stratum corneum, which in turn affects the scalp environment (Lines 347–349)

Q3. If the change in sebum composition has a determining role in AGA why do hair follicles transplanted from the occipital scalp survive? – one would not expect the sebum composition to change as the original sebaceous glands are still in situ. I could understand how the mechanism proposed by the authors may have a modulating effect at a relatively late stage in the pathogenesis of AGA but am less convinced it is a key driver of events.

Response

Chanprapaph et al. reported that UV radiation exposure increases the stratum corneum thickness at the vertex. Even if the occipital scalp along with the original sebaceous glands is transplanted to the AGA region, the vertex is more exposed to UV radiation compared to the occipital region. Therefore, UV radiation exposure may result in sebum oxidation and changes to the stratum corneum thickness, leading to alterations in the water content, TEWL, and subsequently, sebum composition.

Q4. The discussion of the genetics of male AGA could be improved. The Gatherwright twin study (ref 7) only assessed identical twins, although it does not state how monozygosity was determined, and does not provide a heritability estimate. The twin studies by Nyholt et al (2003)and Rexbye et al (2005) both of which included dizygotic as well as monozygotic twins, estimated the heritability of male AGA at between about 80-95%. There are also several more recent studies of AGA genomics than the reference quoted (ref 6), e.g. the most recently published GWAS identified 624 genomic sites associated with AGA (Yap et al Nat Commun. 2018;9:5407).

Response

As reported by Nyholt et al. and Rexbye et al., AGA is significantly affected by genetic factors and male hormones. We do not deny the role of genetic factors; however, we have not discussed them in the paper. As reported by Gatherwright et al., some factors that affect AGA cannot be explained on the basis of genetic differences. Our study showed that factors that affect the scalp environment, such as the microbiome and sebum, may also affect AGA. Furthermore, differences in these factors cannot be explained by genetic differences alone.

Based on your suggestion, we have modified the following sentences:

The sentence “A genetic predisposition and hormonal changes are involved in the pathogenesis of AGA [5]; the former includes polymorphisms of the androgen receptor gene on the X chromosome and the presence of disease-related genes [6].” has been modified to:

“Genetic predisposition and hormonal changes are involved in AGA pathogenesis [5]; for instance, a genome-wide association study identified 624 genomic sites related to AGA [6].”

The sentence “Studies involving identical twins have shown that internal and external factors affect alopecia [7]” has been modified to: “Studies on monozygotic and dizygotic twins have shown that genetic factors are strong [7-8], while studies on monozygotic twins have shown that internal and external factors affect alopecia [9]”.

Reference 6 in the original manuscript has been modified in the revised manuscript, and new references have been added (References 7-8). Because of the addition of new references, the sequence of the references has changed (i.e., references after reference 7 have been moved down by two numbers).

Reviewer 2 Report

The article entitled, "Scalp microbiome and sebum composition in Japanese male individuals with and without androgenetic alopecia", by Suzuki, et al. studies the correlation between scalp microbiome and sebum composition in Japanese males with androgenetic alopecia.  Both variables were compared between alopecia sufferers and same-aged subjects without alopecia. The analysis to correlate each predominated bacterium or fungus species with serum composition is a reasonable approach to the study of microbiome-alopecia pathogenesis.  The conclusions are based on the results and are clearly outlined. I thereafter recommend its publication in your journal.

One small comment: I think the sentence beginning on line 299 will be better in the present tense (since the interest no doubt remains).

Author Response

Reviewer 2

Thank you for your helpful comments and suggestions. We have modified the text accordingly.

The article entitled, "Scalp microbiome and sebum composition in Japanese male individuals with and without androgenetic alopecia", by Suzuki, et al. studies the correlation between scalp microbiome and sebum composition in Japanese males with androgenetic alopecia. Both variables were compared between alopecia sufferers and same-aged subjects without alopecia. The analysis to correlate each predominated bacterium or fungus species with serum composition is a reasonable approach to the study of microbiome-alopecia pathogenesis. The conclusions are based on the results and are clearly outlined. I thereafter recommend its publication in your journal.

One small comment: I think the sentence beginning on line 299 will be better in the present tense (since the interest no doubt remains).

Based on your suggestion, we have modified the text as follows:

The sentence “It was, therefore, of interest to determine whether a particular phylotype was present on the scalps of patients with AGA.” has been modified to:

“It is, therefore, of interest to determine whether a particular phylotype is present on the scalps of AGA patients (Lines 299–300).”
